# Brain Processing of Complex Geometric Forms in a Visual Memory Task Increases P2 Amplitude

**DOI:** 10.3390/brainsci10020114

**Published:** 2020-02-20

**Authors:** Héctor A. Cepeda-Freyre, Gregorio Garcia-Aguilar, Jose R. Eguibar, Carmen Cortes

**Affiliations:** 1Facultad de Psicología, Benemérita Universidad Autónoma de Puebla, Puebla 72000, PUE, Mexico; 2Instituto de Fisiología, Benemérita Universidad Autónoma de Puebla, Puebla 72592, PUE, Mexico

**Keywords:** visual working memory, interference effect, event related potentials

## Abstract

We study the cognitive processing of visual working memory in three different conditions of memory load and configuration change. Altering this features has been shown to alter the brain’s processing in memory tasks. Most studies dealing with this issue have used the verbal-phonological modality. We use complex geometric polygons to assess visual working memory in a modified change detection task. Three different types of backgrounds were used to manipulate memory loading and 18 complex geometric polygons to manipulate stimuli configuration. The goal of our study was to test whether the memory load and configuration affect the correct-recall ratios. We expected that increasing visual items loading and changing configuration of items would induce differences in working memory performance. Brain activity related to the task was assessed through event-related potentials (ERP), during the test phase of each trial. Our results showed that visual items loading and changing of item configuration affect working memory on test phase on ERP component P2, but does not affect performance. However frontal related ERP component—P3—was minimally affected by visual memory loading or configuration changing, supporting that working memory is related to a filtering processing in posterior brain regions.

## 1. Introduction

### 1.1. The Study of Visual Working Memory

Working memory is a cognitive capacity involved in many activities of daily life [1,2,3]. Tracking daily tasks at our jobs, calculating the bill at a restaurant, or making sense of a story being told by a friend are all examples of everyday use of working memory. Working memory as defined by Baddeley and Hitch [4], is a type of dynamic memory, responsible not only for retaining information but also for manipulating it. This manipulation consists of maintaining and updating the items memorized. Visual working memory (hereafter referred to as vWM), as a subsystem of working memory is related to the interactions between vision, attention and behavior [5,6].

Working memory capacity can vary from one person to another [7,8,9,10,11,12]. At least three possible sources of individual variation in working memory have been reported in the past [7]: storage capacity [11], processing efficiency [10], and the ability to maintain relevant information in memory when there’s irrelevant information in the environment [9]. The spatial configuration of items to remember have been reported to influence the memory correct recalls, because this may impose a source of load to memory [13]. More recent studies have further explored this possibility; Schurgin and Flombaum [14] have reported a vWM capacity tolerance to loading effect, particularly when subjects have to remember images of everyday objects. In contrast, Rademaker, Park, Sack, and Tong [15] have reported that the recall process is more vulnerable to loading effect as the time elapsed if the retention interval increases. Also, Chunharas, Rademaker, Sprague, Brady and Serences [16] have reported a beneficial effect of remembered items on correct recalls, when a separation is used. The disposition of items to remember affect also the performance [17]. The configuration of the items, as part of the task objective can also affect performance recall of working memory (WV) [18] Then, some factors that affect visual memory correct recall rate are: position of items, period of retention and configuration of the objects to remember. This information is in agreement with the current notion that WM reflects a separable executive functioning in which the updating of information affects performance of correct recalls, with a strong dependency on the task [19].

It has been reported that differences in performance of vWM research are dependent on these separable roles of cognitive functioning (updating of memory items), mixed with the memory load. For example, high memory load may affect performance when the task demands includes a change of position or a change in configuration of the items to remember. This situation may also be affected if the quantity of items to remember (so called memory load) changes. As this is a common scenario in the real world—for example, when a driver must update in memory the cars that are at left before turning the wheel, as well as be caution about the moving items in the new direction—, then, the study of the interaction between changing conditions that affect the visuo-spatial items to remember is of scientific interest.

Several tasks have been used to measure vWM in the past. The Corsi Block-tapping Task (CBT) [20], is used in the assessment of spatial memory, using sequences of visual locations. Analogous to the recurring digits paradigm, CBT requires the subject to sequentially tap small blocks using a finger. The sequence in which the subject has to tap the blocks is presented by the examiner. Subjects memorize and repeat tapping sequences of increasing complexity, until they cannot longer reproduce them accurately. In CBT paradigm, the effect on working memory is produced by the position of items. The Brooks Matrix Task (BMT) is used to evaluate spatial working memory using mental representations of visual information [21,22]. Subjects are asked to imagine an empty table—a blank square matrix—, and then mentally place numbers in its cells. After placing 8 numbers in the matrix, subject are asked to recall the position of all the numbers. The BMT task also assesses spatial condition of vWM.Another task to assess vWM is the Change Detection Task (CDT), in which subjects are briefly shown a visual arrange of objects, the array is removed from their view, and finally a new array is presented. Subjects are then asked to indicate if they detect any change between the first and second array. The CDT has been used to investigate the number of objects that subjects can remember, as well as the complexity of stimuli features that subjects can deal with statically like shape, color, and orientation [23] and the effect of memory load [17].

Even though spatial tasks usually involve visual stimuli, there is evidence for a dissociation between spatial and object recognition processing in this kind of experimental paradigms [18]. That is, for the CDT, changing one item or more can affect the correct recall when it comes to vWM. However, it is not clear if this effect is because it is change the quantity of items, or because of the complexity of the disposition of items to remember as one and only object. In a recent study it has been demonstrated that the specific positions are updated including the repeating items in an arrange of vWM that allow a higher-order WM representation instead of a global updating process [24]. This is quite important because vWM is dynamic and needs to be reset when moving in a tridimensional space.

### 1.2. Event Related Potentials in the Study of vWM

That said, assuming that item loading, quantity of items to remember and position of items may interfere with vWM correct recall, it is a challenging idea, because other processes besides memory may come into play. In the case of CDT, there are some findings in selective attention paradigms that suggest the co-working of vWM and visual attention (VA). When subjects ought to detect a change in color, shape, or text discontinuities—whether using new pop-out information, frequent information, or a previous template—, the brain electrical activity is reflected in waves of the event-related potential (ERP) components traditionally linked to attention (N1 and P1 [25,26,27,28]), as well as waves related to cognitive manipulation for contextual processing or updating of the stimuli (P3 [29,30,31]). Some other ERP components, like the P2 and N2 waveforms have also been reported to reflect the changes in visual attention related to task demands [32,33,34,35]. When CDT has been used to test working memory, findings are difficult to match with those of attention paradigms. Vogel & Machizawa [36] reported that for feature maintenance in working memory, the early components—around 200 ms—are enhanced in posterior brain regions, and this brain activity raised as the number of stimuli increases during the encoding and maintenance of visual items in memory. This information has been updated by recent research [37,38], which has shifted to focus to the filtering of irrelevant information in the visual memory during the test phase of visual memory. Nasr et al. [37] argued that the negative waves at posterior regions (N200) may reflect the brain activity related to a top-down filtering network for irrelevant features of visual stimuli. Also, the negative frontal wave (N270) may show the way in which visual perception influences the evoking results store in visual memory [38]. Freunberger, Klimesch, Doppelmayr, & Höller [33] also argue for the existence of top-down mechanism related to the P2 component. In their research, by using photographs of living beings and non-living items, they reported a P2 enhancement for incongruent items compared with congruent ones, and a left-hemisphere lateralization of the brain wave also. The authors interpreted the P2 enhancement as a top-down process to match the expected test image, in such a way that incongruent items became deviant from the matching rule and thus elicited a higher task demand.

In other research, Finnigan, O’Connell & Robertson [34], assessed the ERP components in a string of words task and reported the appearance of P2 component in posterior regions related to working memory. These authors used a modified version of Sternberg’s task that relied on words as the main stimuli, which means that they also reported the enhanced wave of P3 component at frontal sites. Remarkably, there were no significant differences for the P3 between young and old adults; on the contrary, the P2 was larger for young adults. P2 was interpreted here as reflecting an encoding of information activity—semantic in this case—which produced a better working memory performance. These results can be understood as the output of attention bias toward relevant stimuli. A well accepted theoretical explanation for this phenomenon relies on ventral and dorsal pathway for visual information processing [39]. According to this, the ventral pathway processes the meaning-what of the information; while the dorsal pathway processes the spatial information of the target. In both cases, the ERP components reflect the encoding, maintenance and recognition of the working memory items. The topographic posterior components of ERP (P2) reflect the sensory early stages of process, and the medium and late stages, representing the anterior topographic ERP components (N2 and P3) indicate the recognition and closure of the process [40]. In the more recent investigation that the authors of this research article are aware of, Kessler & coworkers [24], have reported the effect of updating memoranda over the ERP components. The main affected ERP component when the quantity of items changed, can be observed through enhanced activity in anterior brain regions—Fp1 and Fp2—. However, a posterior activity called contralateral delay activity—CDA—associated with the component for change detection reported by Luck and Vogel N2pc [41] was observed too. Kessler and coworkers argued that this ERP activity is explained because “[i]tems are maintained separately, and can be updated independently, presumably through the operation of item-specific” process (p. 10, [24]). They used a paradigm in which changes had three levels, with two conditions for updating, according to a theoretical manipulation of one-by-one items, called local updating or a global strategy for items processing of visual scanning.

In another research, Talsma & Kok showed that in sensory visual processing areas, the feature related selective attention to stimuli enhanced early components of ERP in a different pattern compared to when the modality or an interfering stimulus is presented within the target [42]. These authors interpret this affect as part of a hierarchical division of attention functioning, in which early components filter the relevant information, and observed in the enhanced activity of sensory neurons. It can be argued at this point that the interplay between memory and attention is a matter of temporal processing. According to Olivers, Peters, Houtkamp, & Roelfsema, the influence of working memory on attention depends of the necessity of an item to recall, so that in can be remember in the form of a template influencing the search processing of the target stimulus. On the other hand, if this template is weak, then the searching of the target is influenced by the accessory memory items involved in the target stimuli [43].

In this case, this interaction between working memory and selective attention could depend on object recognition and closure processes, so that the weaker ambiguity in the target produce a faster neuronal response, because it is recognised and retrieve as a template. On the contrary, the increasing ambiguity in the target stimuli, could generate a slower activation of neurons, because they are recruited in a larger number. This could explained the features of ERP like those reported by Kessler and coworkers [17,24] and Luck & Hillyard [41] will be related to earlier ERP components as those observed by Zhou & Thomas [44] and Quak, Langford, London & Talsma [45]. Since an important source of working memory variations are location, loading and configuration of items conforming a template, and because selective attention could affect the visual searching by locating neuronal resources to the accessory items of visual memory CDT may allow to test it.

That said, in the present report we analyze the effect of memory load along with the items configuration on vWM by changing one item of memory, one item and a background which could be square or an irregular polygon. We studied the behavioral responses of test phase simultaneously with the ERP components of healthy young adults in a modified CDT paradigm. Taken together the results of related previous research, it can be argued for this two options: (a) the CDT paradigm could affect the vWM processing reflected in the recall ratios, and observed in the early configuration of the ERP in posterior regions of the brain, and; (b) if performance is not affected by memory load and configuration, then the correct performance could reflect a selective strategy of attention by recruiting the activity of more neurons in dependence of stimuli configurations changes in early components of ERP in posterior regions We expect to observe the effect in brain activity in visual brain areas during early stages of processing because the geometric form of our stimuli. Following the influence of memory over selective attention statement [43], we expect only a degree of differences between the configuration of our stimuli (blocks), within the same behavioral and brain activity pattern.

## 2. Materials and Methods

This research was approved by Benemerita Universidad Autonoma de Puebla Research and Postgraduate Studies Vicerectory. All participants provided their informed consent.

### 2.1. Participants

Thirty-six undergraduate psychology students (18 women) aged 18–28 (M = 22.4 years, SD = 2.3) participated voluntarily in the study. Subjects were inquired about their handedness, 34 reported being right-handed, and 2 reported being left-handed. No formal tests of manual asymmetry were performed and we relied solely on subjects’ self reports. Subjects reported having normal vision, and reported no familiar background of psycho-pathological disease or neurological alterations. The study followed the Declaration of Helsinki for experiment with humans and was reviewed and approved by the local ethics committee (Faculty of Psychology, 2016). All volunteers signed an informed consent form for voluntary participation and authorized the use of the data generated from their participation. The identity of participants is kept anonymous for the entirety of the study.

### 2.2. Materials

The experimental tasks were programmed and executed in e-Prime 2.0 [46].

Subjects sat approximately 70 cm away from the screen. At this distance, the brightness of the experimental stimuli is on average 30 lx for the Control block, 29 lux for the Square block, and 29 lx for the Irregular block. These measurements were taken in the experimental room with a baseline brightness of 4 lx (when the screen monitor is off), using a digital lx meter model LX1330B.

Electroencephalogram recording was done using an actiCHamp (brainvision) active amplifier (100 KHz sample rate), with a 32-electrode montage according to the international 10-10 system using Cz as a reference [47] on an electrode cap (EasyCap). For the analysis, the EEGs were re-referenced to average reference.

Average impedance of the electrodes remained less than 5 kΩ through the experiments. Digital recording of the electrophysiological data was done through the open source code software PyCorder (Brainvision, Brain Products, Morrisville, NC 27560, USA. Version 1.0.4). The analysis of behavioral data was carried out using the statistical environment R (R Core Team, 2016). Analysis of electrophysiological data was done in the Matlab 2016 environment, through the specialized package “EEGLAB”, version 13.6.5 [48] and the “ERPlab” plugin, version 5.0 [49] together with the statistical environment R. Section 3 includes our results, please see Appendix A for our study available analyzed online data.

### 2.3. Experimental Task

A vWM task, based on change detection task (CDT) was used for the assessment of vWM, as reported in previous research [23,36]. The number of objects per stimulus was chosen to be four based on the findings by Cowan [50,51], while the degree of complexity followed the recommendations of previous investigations [51,52]. The experimental task consisted of 150 trials presented during 4.5 s as maximum duration. Each trial included four images: stimulus onset (300 ms), memory array (1500 ms), retention interval (1500 ms), and test array (1200 ms). All the time intervals were kept unchanged, with the intention of manipulating only the memory load and the stimuli configuration, by changing only one item of the memoranda during the probe. An example train of trials is presented in Figure 1. Visual stimuli consisted of four (out of eighteen) randomly selected irregular black polygons of high complexity, created specifically for the tasks, and based on the stimuli used by Luria, Sessa, Gotler, Jolicoeur and Dell’Acqua [53]. These polygons are shown in Figure 2. Stimuli were presented two times each trial, the first time as part of the memory array, and the second time as part of the test array. In both occasions the four stimuli were arranged in a square pattern.

The vWM task had three experimental conditions, with different levels of visual load and configuration. Although adding visual loading and configuration in the form as a static field, or distortion of the whole image is a common strategy [54,55,56,57], we opted to instead use the outline of a larger polygon as our visual item manipulation. The rationale for this decision was as follows; the loading or configuration alteration included is rather conservative, as we wanted to avoid interfering too much with perceptual processes (e.g., decreasing stimuli identification [58]), this was also the rationale for adding the geometric stimulus as background, rather than foreground. On the other hand, we also wanted to ensure that subjects could not avoid looking at the background stimulus, to achieve this we made sure that all the polygons were in direct contact with the geometric background stimulus. Regarding the usage of a polygon as background stimulus, we based on previous evidence that recall performance may depend on the task-relevant feature to memorize [52], and thus we attempted to keep the same “feature domain” of the task-relevant polygons (i.e., the only differentiating feature is shape). Also, Talsma, Wijers, Klaver & Mulders [59], reported an effect of polygons ipsilateraly or contralaterally presented over ERPs in posterior regions of the brain. Then, not only the item number is reported to affect the correct recall, but also location and identity of items features. So, we also attempted to manipulate the spatial location of items by presenting it in the corners of the test stimuli. The first condition, labeled control, shows only the task-relevant polygons. The second condition, labeled square, had a gray frame forming a square, set up behind and coinciding with each polygon as a corner. The third condition, labeled irregular condition, had presented an irregular polygon as a frame behind the polygons. Figure 3 shows an example of these experimental variations of stimuli. These conditions were programmed to form three independent blocks presented randomly in each experimental session. Fifty percent of trials were set up to have a change in test stimuli. Also, we varied the configuration of the polygons except for the background polygon.

A 32 electrode placement was done once the subjects were sitting comfortably, accordingly to the 10-20 system and using the Easycap by brainvision. Participants received instructions regarding experimental tasks, and had some sample trials to practice, allowing the subjects to get used to the task. Once each participant confirmed understanding the task, the first experimental block started. Each experimental block lasted approximately 11 min. Between each block, a break was allowed. A numeric keypad with only two keys (1 and 3) was set for behavioral responses. Participants were instructed to respond as quickly and accurately as possible by pressing a left key with the left thumb if they detected a change, or right key with the right thumb if they did not recognize a difference between the memory array and test array on each trial. EEG was recorded continuously and synchronized to each block task performance.

### 2.4. Data Analysis

Participants executed the three experimental conditions of the vWM task, as experimental blocks. Hence, the experimental design was a block-repeated measurement design. Blocks were presented in a counterbalanced manner, according to a Latin Square design, in six possible configurations. Control condition as block 1, the square condition as block 2, and the irregular condition correspond to block 3. The configuration according to which participants carried out the experimental session had been set up through a random sample without replacement procedure, resulting in a three-men and three-women allocation for six configurations. The first part of the analysis aimed to test sensitivity and response bias of participants’ responses through signal detection theory measures, based upon the procedure described by Stanislaw & Todorov [60]. Afterwards, for behavioral data, the Friedman test was used. The ERP analysis consisted of repeated measures of 3 factors (Block X Hemisphere X Scalp Zone) Analysis of Variance test (ANOVA). For the factors that showed a significant effect, Bonferroni-corrected *post-hoc t*-tests were performed. The interaction between ERP components and behavioral variables was tested through a linear mixed effects model. The residuals for such model were assessed to verify the assumptions of normality and homoscedasticity.For all ERP analysis, we only report the test stimuli of the paradigm without differentiate between changed/not-changed test stimuli; that is, ERP activity reported here correspond to only correct recalls. We excluded the visual processing and retention period.

#### ERP Analysis

The ERP analysis was performed using a standard procedure (Handy, 2005; Luck, 2005d). The preprocessing of the raw EEG signal was carried out using the ERPlab toolbox. The steps followed were as follows; Step 1. A high-pass of 0.1 Hz and a low-pass of 50 Hz [61] with a Finite Response Filter in the order of 4092 were used to filter the data (i.e., remove line noise and slow waves). Step 2. We split the EEG data into epochs, only epochs corresponding to the test array were included in the analysis. Epochs started 200 ms before test array onset, this time period was used for baseline correction. Epochs ended 857 ms after test array onset, this upper limit was equivalent to the average response time of the sample. Step 3. The peak-to-peak method was used for detecting and rejecting artifacts (the criteria was a 100 μV peak-to-peak difference within a 200 ms window). This method eliminated most of the epochs in which blinking was present. Visual inspection [49,62] was used as a complement to peak-to-peak detection. As a result of this step, any epochs in which subjects were blinking (e.g., as a consequence of screen brightness), or had excessive eye moment were removed from analysis. The mean number of epochs used in the analysis per subject was 118.61 (±32.43 SD) for the Control condition, 116.94 (±31.61 SD) for the Square condition, and 119.52 (±32.04 SD) for the Irregular condition. A Friedman test indicated no significant differences in the number of approved epochs ($\chi^{2}$ = 2.46, df = 2, *p* = 0.29). Step 4. After peak-to-peak rejection, the EEGs were re-referenced to average reference.

To determine the time windows of ERP components, visual inspection of the grand-average waveforms was performed. Time windows were selected in such way that the peak was at the middle of the time window, upwards and downwards tails of the waveform were captured, and no overlapping with other component time windows occurred. Additionally, we followed Luck’s [63] description of common time windows for the components. As a result, the selected time windows were 200–260 ms for the P2 component, and 300–500 ms for the P3 component. Although we expected to observed a N2 deflection, this was not clearly observed on the grand-average waveform, and thus we were not able to define a time window, resulting in its exclusion from the analysis.

Rather than analyzing electrodes individually, they were grouped accordingly to their respective scalp zones, as shown in Figure 4. Voltages were averaged to reflect scalp-zone activity.

## 3. Results

### 3.1. Behavioral Results

#### 3.1.1. Bias

Response bias was tested for normality using Shapiro-Wilk tests. The Control block passed the normality test (*W* = 0.98, *p* = 0.75), but the other two blocks failed the test: Square block (*W* = 0.93, *p* = 0.04) Irregular condition (*W* = 0.93, *p* = 0.027). Subjects showed a tendency to respond “NO” in the three blocks. For the Control block the median of *c* was −0.73 (±0.43 Median Absolute Deviation, MAD); in the Square block, the median was −0.68 (±0.36 MAD), while in the Irregular block the median was −0.75 (±0.43 MAD). Figure 5 shows the bias of the participants.

#### 3.1.2. Accuracy and Response Times

Normality was also assessed for the accuracy of the responses and response times using Shapiro-Wilk normality tests. These results are depicted in Figure 6a. Other blocks did not follow a normal distribution. Hence, we decided to analyze behavioral results using no-parametric Friedman tests. The median response time in the control block was 849.57 (±75.52 Median Absolute Deviation, MAD) milliseconds; for the square block, the median response time was 871.57 (±41.06 MAD); and for the irregular block the median response time was 877.42 (±67.31 MAD). A Friedman tests did not show significant differences between blocks ($\chi^{2}$ = 0.16, df = 2, *p* = 0.92). Figure 6b depicts the response times of the participants.

The median for response accuracy in the control block was 0.55 (±0. 09 MAD); for the square block, the median of correct responses was 0.54 (±0.11 MAD); for irregular block median of correct responses was 0.53 (±0.09 MAD). Friedman test analysis for correct responses did not show differences between blocks ($\chi^{2}$ = 1.44, df = 2, *p* = 0.48). Figure 6 depicts the correct responses emitted by the participants.

### 3.2. ERP Results

Task-related brain activity was higher in occipital regions, spreading out to parietal and temporal regions, while in the frontal region showed the less electrophysiological activity. Figure 7 depicts the ERP components related to the task. Even though early components P1 and N1 reflected the differences between attentive processes related to the task [25,64,65], later component P2 at occipital zones reflected the main brain activity for the task demand.

### 3.3. P2 Analysis

Mean amplitudes during the P2 time window were analyzed as the dependent variables for these results. This time window can be observed in the context of the ERP waveform in Figure 8.

A repeated measures ANOVA with the voltage of the P2 (mean amplitude in μV) as a dependent variable was carried out and the factors Block x Zone x Hemisphere as independent variables. This analysis indicated a significant effect of the Block ($F_{2,70}$ = 31.19, p< 0.001, $\eta_{G}^{2}$ = 0.004), the Zone ($F_{3,105}$ = 47.55, p< 0.001, $\eta_{G}^{2}$ = 0.44) and the Interactions Block x Zone ($F_{3,105}$= 15.51, p<0.001, $\eta_{G}^{2}$ = 0.022) and Zone x Hemisphere ($F_{6,210}$ = 9.40, p< 0.001, $\eta_{G}^{2}$ = 0.007). No significant effect was found for the variable Hemisphere, nor of the interactions Block x Hemisphere and Block x Zone x Hemisphere (p< 0.05). Detailed results can be observed in Table 1.

*Post-hoc* analysis revealed significant differences between the Control Block and the Square Block in the occipital zone (df = 35, *t* = 6.31, p< 0.001, *d* = 1.05). In addition, the analysis revealed significant differences between the Control Block and the Irregular Block in the frontal zones (df = 35, *t* = −4.16, p< 0.01, *d* = 0.69), temporal (df = 35, *t* = 3.81, p< 0.001, *d* = 0.63), and occipital (df = 35, *t* = 6.07, p< 0.001, *d* = 1.01). No significant differences were found between the Irregular and Square blocks in any of the analyzed brain regions. The results of the experimental blocks comparisons are shown in Figure 9.

With respect to areas of the scalp, the *post-hoc* analysis found significant differences for the Control Block between the Occipital and Frontal zones (*t* = 6.08, p< 0.001, *d* = 1.01), Temporal (*t* = 6.41, p< 0.001, *d* = 1.06), and Parietal (*t* = 3.59, p< 0.001, *d* = 0.59). We also found significant differences between Parietal and Frontal zones (*t* = 5.42, p< 0.001, *d* = 0.90), but not between the Parietal and Temporal or Frontal and Temporal zones (p> 0.05). Within the Square block there were found significant differences between the Occipital zone and the Frontal (*t* = 8.52, p< 0.001, *d* = 1.42), Temporal (*t* = 7.54, p< 0.001, *d* = 1.25), and Parietal zones (*t* = 7.33, p< 0.001, *d* = 1.22), as well as between the Parietal and Frontal zones (*t* = 6.58, p< 0.001, *d* = 1.09), but no significant differences were found between the Temporal and Frontal zones (p> 0.05). Finally, in the Irregular block, significant differences were found between the Occipital zone and Frontal zones (*t* = 7.87, p< 0.001, *d* = 1.31), Temporal (*t* = 8.14, p< 0.001, *d* = 1.35), and Parietal (*t* = 7.37, p< 0.001, *d* = 1.22). The Frontal zone showed significant differences compared to the Parietal (*t* = 5.81, p< 0.001, *d* = 0.96) and Temporal zones (*t* = 5.35, p< 0.001, *d* = 0.89). However, no significant differences were found between the Parietal and Temporal zones (p> 0.05). The results of the scalp zones comparisons are shown in Figure 9.

### 3.4. P3 Analysis

A repeated measures ANOVA showed a significant effect of the Scalp Zone (F3,105=20.43,$p<0.001,\eta_{G}^{2}=0.24$), the Hemisphere ($F1.35=10.92,p<0.01,\eta_{G}^{2}=0.02$), and the interaction Zone x Hemisphere ($F3,105=8.70,p<0.01,\eta_{G}^{2}=0.01$). However, no significant effect was found between blocks (p>0.05). Detailed results are shown in Table 2.

The post-hoc analysis showed significant differences for the left hemisphere between the Frontal and the Occipital (t=5.53,p<0.001,d=0.92), Parietal (t=6.42,p<0.001,d=1.07), and Temporal zones (t=3.35,p<0.01,d=0.55); also, the analysis showed significant differences between the Temporal and the Occipital zones (t=4.88,p<0.001,d=0.81) and Parietal zone (t=3.94,p<0.002,d=0.65). No significant differences were found between the Occipital and Parietal zones (p>0.05). For the right hemisphere significant differences were found between the Frontal area and the Occipital zones (t=4.26,p<0.001,d=0.71), Temporal (t=5.00,p<0.001,d=0.83), and Parietal (t=5.55,p<0.001,d=0.92). These results are depicted in Figure 10.

Between hemispheres, significant differences were found in the Temporal (t=5.01,p<0.001,d=0.83) and Parietal areas(t=4.61,p<0.001,d=0.76), but not in the Frontal or Occipital zones (p>0.05).These results are depicted in Figure 11.

### 3.5. Behavioral and ERP Interactions

A linear mixed model regression analysis was carried out with P2 voltage as dependent variable. Experimental block, accuracy, and reaction time were tested as predictors. This model showed that the best predictor of the amplitude of the P2 was the scalp zone (χ2(2)=518.27,p<0.001). The Block version of the vWM was also a significant predictor (χ2(2)=7.12,p<0.05). Compared with the Control block, the Square block enhanced the voltage of P2 by 0.45 μV (SE±0.21); also, the Irregular block increased the voltage of P2 component by 0.51 μV (SE±0.21) compared to Control block. The Reaction Time was not a significant predictor of P2 voltage (χ2(1)=1.5,p=0.218).

### 3.6. Match-Mismatch Effects on P2 Amplitude

We also analyzed the differences in voltage between Match (No change in stimulus) and Mismatch (Change in stimulus) trials. We selected only correct trials, and performed a repeated measures ANOVA. Similarly to the analysis for P2 voltage, we used P2’s mean amplitude as dependent variable and the factors Block x Scalp zone x Hemisphere x Condition (Match/Mismatch) as independent variables. We found a significant effect of the Match/Mismatch condition ($F_{1,35}$ = 5.26, p< 0.05, $\eta_{G}^{2}$ = 0.0003), as well as a significant effect of the interaction Condition x Scalp Zone ($F_{3,105}$ = 4.42, p< 0.05, $\eta_{G}^{2}$ = 0.002). The Match/Mismatch effect did not interact significantly with any other factor.

In the *post-hoc* analysis, a Wilcoxon rank sum test with bonferroni correction showed significant differences in the P2 amplitude of Match and Mismatch trials with regard to the Frontal zone (V=8922,p<0.01) and the Occipital zone (V=14067,p<0.01), the rest of the scalp zones showed no significant differences between onditions of trials. These comparisons can be observed in Figure 12.

## 4. Discussion

In this study, we present evidence that visual item loading as well as item configuration in a vWM task may increase the brain activity at occipital regions 200 ms after stimulus onset, during the test period. The electrophysiological activity observed was found mainly in occipital and posterior regions, which agrees with previous suggestions regarding the role of sensory processing in which interact the working memory and the selective attention processing, activating cortical posterior regions. There is evidence that occipital and parietal brain areas are responsible for vWM processing [66,67] and attentional visual modality of filtering [42].

However, the experimental manipulation showed no effect on behavioral measurements of vWM, that is, the visual item loading and item configuration did not increase nor diminish neither the precision of correct responses, reaction time, sensitivity, nor bias of responses. The lack of behavioral effect in vWM task in our research agrees with the hypothesis for a cognitive process that regulates visual processing, allowing the subjects to focus on task-relevant elements of the stimulus [68,69], which suggests that it is the task demands [18] and the context [70] that influence the strategy to remember items, in a double way of memory effect presentation: interacting with attention or not interacting [43]. Our interpretation is that the configuration variation and complexity of our CDT stimuli kept same cognitive effort between blocks and between subjects, and produced almost equal recall ratios, because it represented a twofold process: first, the retention as a block or template that spreads over the test stimuli in a processing of searching for equal/different items.

Our behavioral results are not novel however. It has been shown in the past that vWM is tolerant to visual item loading and configuration change [14]. This effect is not exclusive to visual stimuli [71], so it is possible that it is a general characteristic of working memory, not only of visual modality. Furthermore, the lack of behavioral effect of visual loading and item configuration in our experiment can be explained in light of the ‘fix resolution slot model’, according to which, vWM stores discrete information in clusters (‘slots’), that are not affected by memory load. Instead, this ‘slots’ allows vWM to recovery information by filtering the attention’s processing, in dependence to the task objective [6]. Hence, since our paradigm did not change the number of complex geometric forms between blocks but rather added a background geometric one, and modify only one item of the probe array, brain effort could be increased by adding a selection process, without affecting behavioral responses. This effect maybe also the result of a depth separation in constant form of the memorized items (memoranda) [13,16] in our study. The behavioral results of Talsma and collaborators [59] contribute to our explanation, because they found that location is also memorize even if the task’s demands requires only the identity of the items. That said, it could be that for the configuration of complex geometric forms used in our study, a global strategy of vWM would suffice for the cognitive updating function to achieve correct responses, and this strategy could represent a ERP different effect that we will discuss later. On the other hand, this may interpretation account for the subjects’ tendency in our experiment to bias toward answering for no change detected. That is, a conservative tendency for answering no match with the global strategy of WM, making global selection strategy the best one for the task, as long as this strategy allows for the separation of constant forms of background from changing corners in which the geometric complex forms were placed within the stimuli. In this way, the memory item changed in the test array, could be the target of an attention processing. However, further research is necessary to confirm this assertion.

In the ERP analysis of this study, we observed ERP activity primarily on Occipital zones, particularly a deflection which corresponds to the component P2. P1 and N1 deflections were also observed in occipital zones in a constant manner, and with no statistical differences between conditions; and also we observed a minor P3-like deflection. This P3-like deflection was not significantly affected by the experimental manipulation, except for the control block. Our interpretation of this difference relies on a probably more general working memory processing influencing the recalling of configuration in the test stimuli. That said, it could be that this difference reflect the memory configuration like a template to be remember, so that WM affects the selection processing of sensory item. This is in line with one function attributed to memory by Olivers and collaborators [43].

Regarding the P2 waveform, this component has been associated with working memory before [72,73] in similar CDT paradigms using numbers, strings of words, and landscape photographs. Lefebvre, Mrachand, Eskes & Connolly [73], reported a P2 and a slow positive wave (SPW) in posterior regions related to a span digit backward task. However, since we did not find any statistically significant difference in overt behavior between task blocks, and P2 voltage was not related to individual performance, our data can not be related with correct recalls, but maybe with a sensory modality [42,62], or with a filtering process dealing with memory items [43]. Namely, the P2 component may reflect a sensory filtering processing. This interpretation of P2 does not explain neither the voltage increasing according to visual loading between blocks nor the changed configuration on one corner of the stimuli however. An alternative explanation could be that P2 reflects a “top-down” processing effect [33,74] for maintaining the memorized items as one whole object, and for preventing it from contaminating surrounding stimuli. Such interpretation could also explain why we observed a voltage increasing in that component, related to the complexity of the background geometric form, in the absence of behavioral differences. It could be that the P2 reflects a top-down process for focusing on stimuli characteristics previously identified by the subjects, as a process completely separated from recall. Following this logic, the lack of a behavioral effect could be the result of the top-down filtering reflected by the P2 component, allowing the subjects to avoid diminishing performance even in the presence of visual items loading or items change configuration. And because one geometric background was added to the probe stimuli, the maintaining and filtering functions reflected by P2 enhanced voltage differences between blocks. We are opting for a global strategy of the WM, in which the sensory components reflect the cognitive effort for deciphering at once between change and not change of the probe array. In this account of our ERP results, there is another effect of WM, the one of filtering the information by measuring the correspondence of the probe stimuli with the previous one [19]. In this case, the function of WM is not updating, but biasing cognitive processing of WM to achieve the task goal, by balancing the incoming information [43].

Another explanation can be found in a study by Henare, Buckley and Corballis [75], in which they argue that the ERP waveform can be influenced more by distractors than by target stimuli. In the same line, a study by Gaspar, Christie, Prime, Jolicoeur, & McDonald [76], also showed distractor-related changes in ERP waves, and although these changes do not generate wave-forms similar to ours, they appear in close temporal proximity to the P2 component. These authors argue that the ERP changes they observed reflected a suppression mechanism used to process distractor items. We interpret this as the filtering processing for achieving the task goal. Hence, prior knowledge of the context of the task would function as a regulator for attending a new stimulation within the same context [77]. This interpretation suggest that spatial location of item is also involved in the recall process, but, because we did not present the probe stimuli laterally as did Talsma and collaborators [59], we observed a posterior effect in both hemispheres. In other words, given the characteristics of our stimuli, it is possible that the processing in this task is more influenced by the stimuli configuration features than by the added item as background. Then, the general characteristics of stimuli could be regulating the task performance through a balance between the incoming information and the task goal, probably reflected at the sensory level of primary and secondary visual areas. This effect could be incremented due to the habituation to the task, creating a bias in the responses of the subjects.

Following our research goal, it appears that sensory filtering of vWM is done in posterior brain regions, when the stimuli features involved complex representation. Our results agrees partially with those reported by Talsma and collaborators [59], because we observed the constant appearance of early P1 and N2 ERP components. The main effect of our experimental manipulations was observed later in time processing, at the latency of P2 component. This could mean that because our task involved 4 items, the memory load affected later in time the brain activity, which differed from the memory load of one or two items in the visual field. On the other hand, this effect is greater when there is no change (match) in the probe stimuli, than when there is a change (no-match). However, we are caution about this statement, because as has been showed in previous investigations, the P3 central and frontal components’ voltage could be diminish when there exist increasing memory load [78].

In our study, because subjects could be biased by instruction, then the primary and secondary cortices function to maintaining and filtering the visual data with loading and/or changing configuration features. Along with this, the gray and white colors used in our stimuli could enhanced the P2 voltage. In the same vein, this hypothetical function of global processing of filtering by WM could be the mixed of attention and memory processing that affect the physiological data of our experiment. However, in the previous similar research by Kessler and coworkers (Figure 5 right below corner [24]), we found a similar component, close to the time window of P2 in our experiment, that those authors neither explained nor report. Also, in the previous research related with geometric forms as stimuli [59], was observed a P2 wave, with no statistical differences between one or two stimuli to recall in posterior zones. We interpret this component as representing a kind of labour division of WM, with the updating of low resolution but highly cognitive effort demand items of memory carried out by frontal sites; while global strategy, subserving filtering of attending and retained stimuli features process by posterior regions (whether or not interacting with frontal activity), all of which depend on task demands. These demands could be related to memory load and spatial configuration. Since our vWM task enhanced the functioning of posterior brain sites, it can be used to test the hypothetical statement of a different interplay between memory and attention, as we interpret this way the cognitive effort demands by our task [43]. But there are needed more studies to prove this.

### Limitations

Due to the presentation method and the lack of a jitter phase, subjects had more propensity to habituation. This could have influenced the attention levels, performance, or both. The response buttons remained the same through the experiment, hence an opportunity for further counter-balancing was missed, as the response mappings left and right buttons were never inverted. It is also worth noting that there was a strong bias in subjects to declare that no change was present. This could be interpreted as a generalized difficulty in remembering the polygons, or an artifact of a preference to push a button using the right hand.

Finally, the absence of behavioral changes should be noted. It could be that the vWM items loading between blocks, or the changed configuration or not in probe stimuli was not sufficient to generate a behavioral change. While it is possible that increasing the features changes in test array (e.g., by replacing it with a static-like pattern overlay on the whole screen) could produce behavioral effects, it is also likely that the modality we selected is simply unrelated to vWM processing, and more associated with attention processing, so that our hypothesized behavioral change would be due to an attention processing instead. In conclusion, we report that increasing the quantity of memory load and configuration in a CDT, may involve the interaction of attention processing observe in the brain dynamics of posterior regions, in a particular enhancement of P2 component of the ERP.

## Figures and Tables

**Figure 1 brainsci-10-00114-f001:**
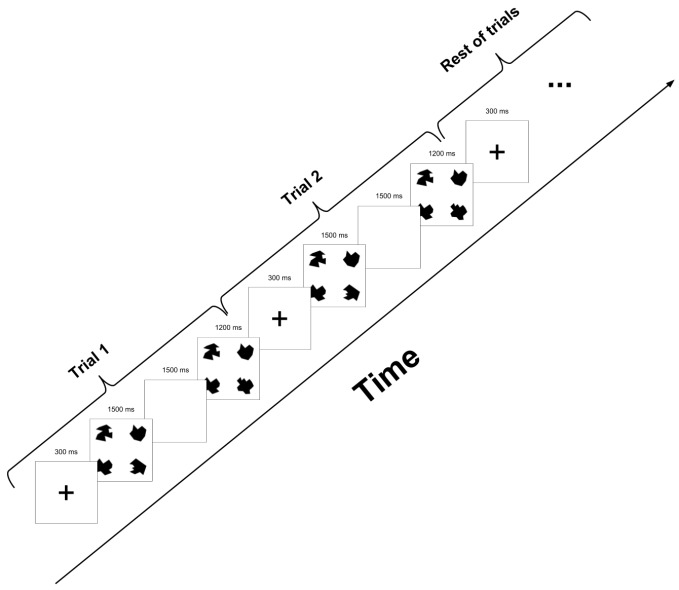
Example train of the task used in the experiment. Each trial was composed of 4 periods: Stimulus onset—a cross-hair was displayed on the screen for 300 ms—, Memory array—Stimulus is shown for 1500 ms—, Retention interval—screen remains blank for 1500 ms—, and Test array—a new stimulus is shown and the subject has to decide whether it is the same from the memory array or not—. After finishing the 4 periods of a trial, a new trial begins.

**Figure 2 brainsci-10-00114-f002:**
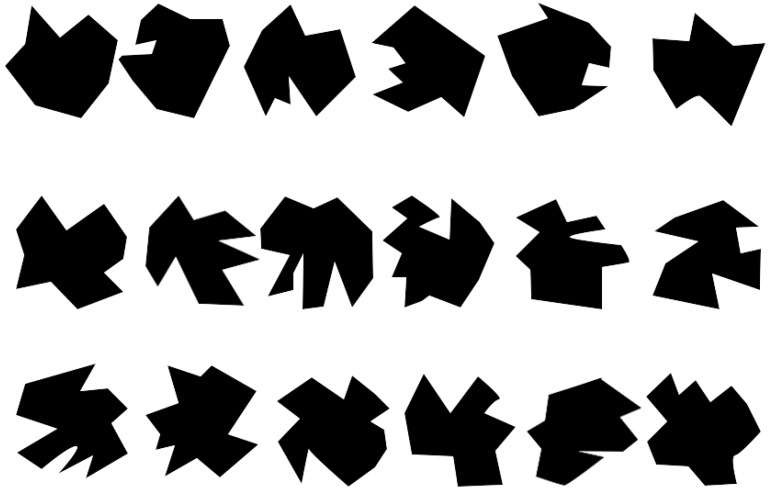
Pool of polygons from which the 4 polygons to create each stimulus were pulled. The polygons used for the arrays were generated previous to the experiment. The arrays were formed taking random polygons without replacement from this polygon pool. All arrays were verified to about repeated combinations of polygons. In total, 450 unique combinations were used to create the arrays.

**Figure 3 brainsci-10-00114-f003:**
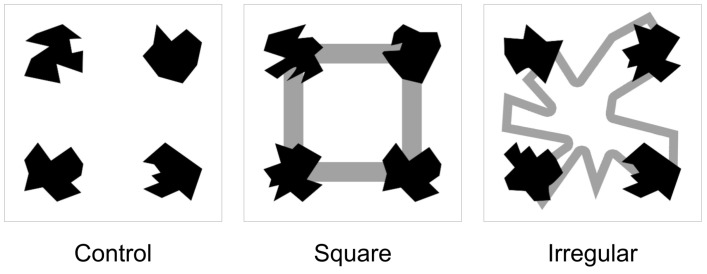
Experimental manipulations in each experimental block. Every experimental block was carried out as described in Figure 1. In the Control condition, the arrays were shown by themselves, with no additional visual stimulation. In the Square condition, four lines forming the perimeter of a square were added to the background. In the Irregular condition, lines forming the perimeter of an irregular polygon were added to the background. The background for the Irregular condition remained unchanged through the experiments, interestingly, some subjects commented after the experiment that they perceived it at changing.

**Figure 4 brainsci-10-00114-f004:**
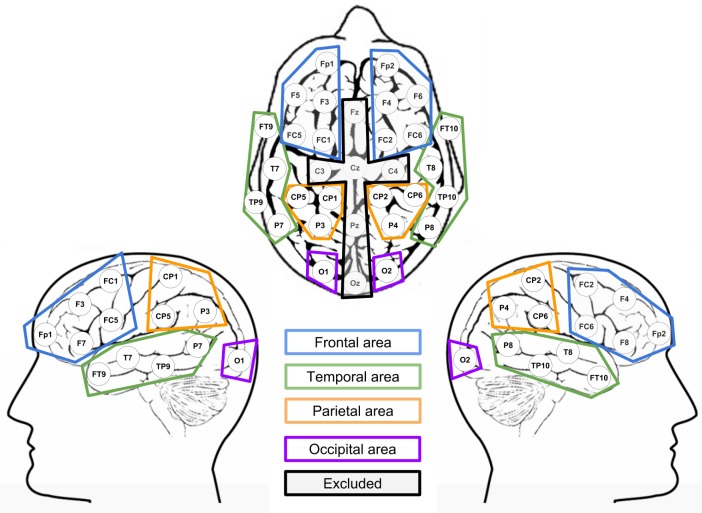
Scalp-zone grouping of the electrodes The corresponding scalp zones and hemispheres for the analysis were set up in the following way: Left frontal zone: Fp1, F3, F7, FC1, FC5; Right frontal zone Fp2, F4, F8, FC2, FC6; left temporal zone: FT9, T7, TP9, P7; right temporal zone: FT10, T8, TP10, P9; left parietal zone: CP5, CP1, P3; right parietal zone: CP6, CP2, P4; left occipital zone: 01; right occipital zone: 02.

**Figure 5 brainsci-10-00114-f005:**
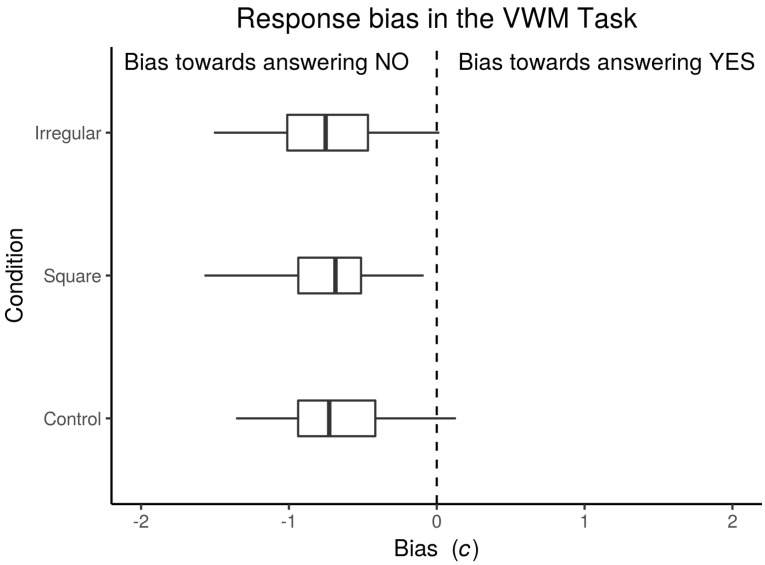
Response bias. Boxplot of the bias measured from the subjects (*n* = 36) during the vWM Task. *c* was used as an index for bias, according to Signal Detection Theory [60]. It should be noted that even though there was a bias towards answering by declaring no change was detected, not every subject showed this bias.

**Figure 6 brainsci-10-00114-f006:**
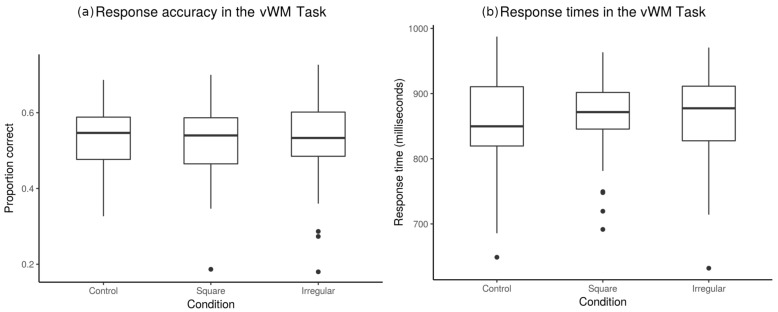
Response accuracy. (**a**) Boxplot showing the accuracy measured from the subjects during the task. Accuracy is simply the number of correct responses divided by the total number of trials. No significant differences were found between the conditions. (**b**) Boxplot showing the response times of the subjects during the tasks. No significant differences were found between the conditions.

**Figure 7 brainsci-10-00114-f007:**
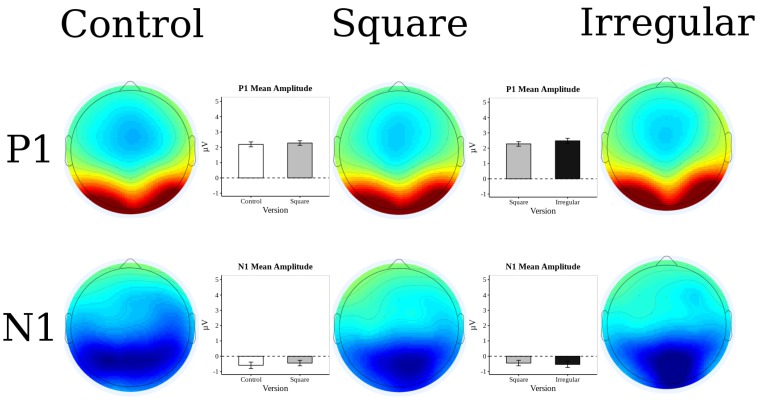
Topographic distribution of event-related activity. Values plotted are mean amplitudes within the time-frames of the respective components. Zones shown in more red have higher positive amplitudes. Zones shown in dark blue have more negative amplitudes. Zones shown in teal are close to 0 amplitude.

**Figure 8 brainsci-10-00114-f008:**
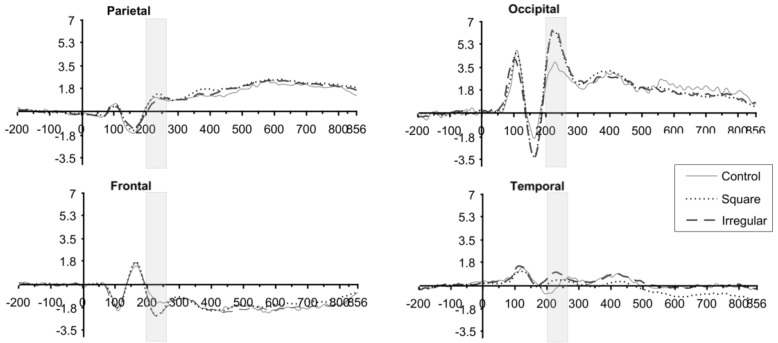
ERP Waveform during the different experimental blocks. ERP waveform observed during the “test” array presentation. The electrodes averaged for every scalp zone are shown in Figure 4. Waveforms are sample averages. The type of line indicates the experimental manipulation from which each waveform was extracted.

**Figure 9 brainsci-10-00114-f009:**
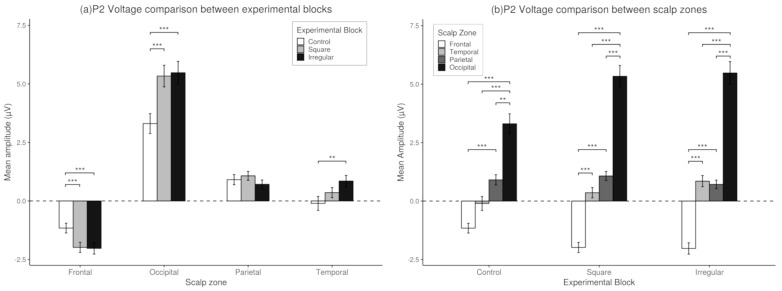
(**a**) *Post-hoc* comparisons for the P2 Component. Experimental block differences. Mean amplitude registered in the different experimental blocks. *Post-hoc* comparisons within scalp zones using Bonferroni’s correction on paired *t*-tests. Values plotted are means ± standard errors. (**b**) Post-Hoc comparisons for the P2 Component. Scalp zone differences. Mean Amplitude registered in the different scalp zones. *Post-hoc* comparisons within experimental block using Bonferroni’s correction on paired *t*-tests. Values plotted are means ± standard errors Stars indicate significance levels: * = p<0.05, ** = p<0.01, *** = p<0.001.

**Figure 10 brainsci-10-00114-f010:**
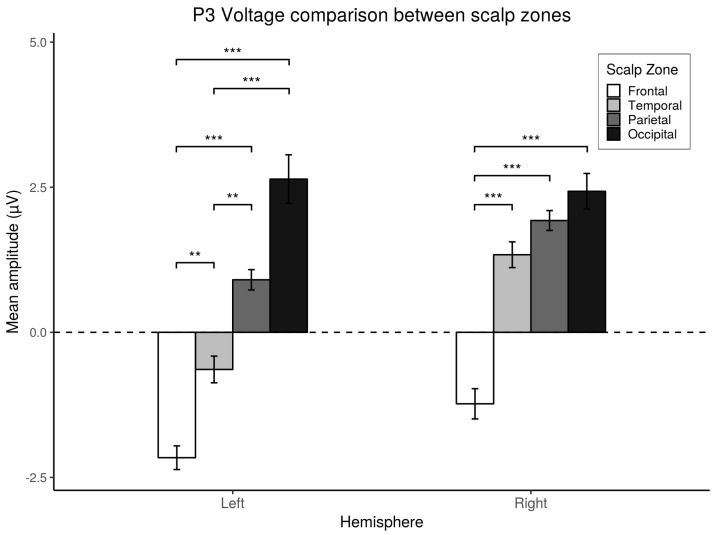
*Post-hoc* comparisons. Scalp zone differences for the P3 component Mean amplitude registered in the different scalp zones. *Post-hoc* comparisons within hemispheres using Bonferroni’s correction on paired *t*-tests. Values plotted are means ± standard errors. Stars indicate significance levels: ** = p<0.01, *** = p<0.001.

**Figure 11 brainsci-10-00114-f011:**
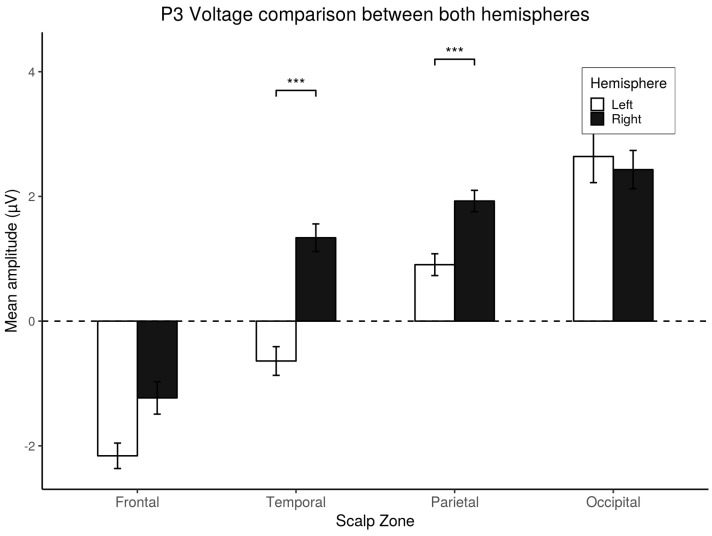
*Post-hoc* comparisons between hemisphere showed differences for the P3 component. Mean amplitude registered in both hemispheres. *Post-hoc* comparisons within scalp zones using Bonferroni’s correction on paired *t*-tests. Values plotted are means ± standard errors. Stars indicate significance levels: *** = p<0.001.

**Figure 12 brainsci-10-00114-f012:**
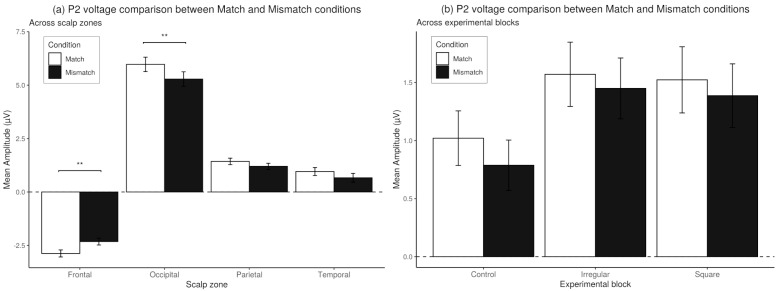
(**a**) *Post-hoc* comparisons for the P2 Mean amplitude between match and mismatch trials. *Post-hoc* comparisons within scalp zones using Bonferroni’s correction on Wilcoxon signed rank test. Values plotted are means ± standard errors. (**b**) Mean amplitude of the P2 Component on match and mismatch trials, across experimental blocks. As the interaction between the factors Block X Match/Mismatch condition was not significant, no *post-hoc* tests were performed. Stars indicate significance levels: ** = p<0.01.

**Table 1 brainsci-10-00114-t001:** Repeated measures ANOVA on the mean amplitude (μV) of the P2 component during the test period.

Effect	Ss	Ms	df(Factor)	df(Error)	*F*	*p*	$\eta_{G}^{2}$
Block	45.83	22.915	2	70	31.19	<0.001	0.004
Zone	4654	1551.2	3	105	47.55	<0.001	0.444
Hemisphere	14.7	14.69	1	35	1.14	0.291	0.001
Block*Zone	238.5	39.75	6	210	15.51	<0.001	0.022
Block*Hemisphere	2.57	1.285	2	70	0.63	0.545	<0.001
Zone*Hemisphere	78.07	26.023	3	105	9.40	<0.001	0.007
Block*Zone*Hemisphere	2.16	0.360	6	210	1.02	0.383	<0.001

*p* values shown were generated using Greenhouse-Geisser correction.

**Table 2 brainsci-10-00114-t002:** Repeated measures ANOVA on the mean amplitude (μV) of the P3 component during the test period.

Effect	Ss	Ms	df(Factor)	df(Error)	*F*	*p*	$\eta_{G}^{2}$
Block	0.56	0.28	2	70	0.36	0.694	<0.001
Zone	2102	700.8	3	105	20.44	<0.001	0.242
Hemisphere	186.4	186.41	1	35	10.92	0.002	0.021
Block*Zone	16.4	2.73	6	210	0.776	0.535	0.001
Block*Hemisphere	1.88	0.9418	2	70	0.437	0.625	<0.001
Zone*Hemisphere	129.8	43.28	3	105	8.705	0.001	0.014
Block*Zone*Hemisphere	5.99	0.99	6	210	1.908	0.127	<0.001

*p* values shown were generated using Greenhouse-Geisser correction.

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
