# Peer review of "Brain Processing of Complex Geometric Forms in a Visual Memory Task Increases P2 Amplitude"

_brainsci, 2020, doi:10.3390/brainsci10020114_

Round 1

Reviewer 1 Report

The authors responded sufficiently to all my comments.

Line 146: Typo in Pos_t_graduate studies

Author Response

Thank you very much for your observations. We appreciate it all. We think they improved the meaning of our words in the document.

As you pointed out, there exist a typo in line 146 for the word Pos_t_graduate studies, we have changed it.

In the acknowledge section, we thank for the support of Pos_t_graduet studies Vicerrectory. We also corrected this typo.

Reviewer 2 Report

Main issues:

It is unclear what the study adds to the literature. More specifically, the structure of the introduction should improve, particularly the latter part. It should become clearer what the gap in literature is that the authors want to address, and why. It is hard to follow where the hypothesis comes from, and what question is asked in the first place. “Our hypothesis is that if vWM performance is affected by visual memory configuration (the disposition of items in memoranda), then by exposing subjects to the same memory task under visual memory configuration change”: It is not clear to me why the authors argue that the configuration changes, when the three conditions differ in the complexity of the background. Please clarify through literature what the background change can do with visual processing. If the authors want to test the effect of configuration, the study design should be different. It is not until late in the method section (ERP data analysis) that the authors explain that they are interested in the test phase of working memory. It makes a lot of difference in cognitive processing memorizing and testing changes. The ERP literature the authors refer to is a mix of the two processes. Earlier work should be referred to that looked at the same kind of stimuli during a similar task (e.g. Talsma et al. 2001), finding similar ERPs. The authors should discuss attentional load theory (Lavie) and Olivers (e.g. Olivers 2011, Peters et al. 2009) on how irrelevant information is processed during working memory tasks. The authors should separately analyze matching and mismatching trials. The P2 is sensitive to those differences in working memory tasks (e.g. Talsma et al 2001)

Minor issues:

Abstract

Explain when ERPs were recorded, during memorization or test. “other ERP component” – syntax error

Intro:

It is often unclear whether the authors refer to the period between memorization and test, or after test “change detection phase”. The authors seem to switch back and forth. E.g. Vogel & Machizawa (CDA) refer to the first phase, while Gao et al refer to the latter phase. It is not until line 240 that it becomes clear that the authors are interested in the test period. Syntax line 57: “Then BMT task..” line 53 “in CBT paradigm”. line 53 “items The” line 58 “The Change Detection Task” -> “the Change …” 108: “bot cases” Paragraph starting at line 123 is quite unclear. Please rephrase thoroughly Paragraph starting at line 134. The question is unclear. What is the open question that this study focuses on and why? Please discard sentence about language in 142. It makes no sense: 1) the authors did not test whether language is not involved, so it cannot be claimed that it is “unbiased”, b) do not claim there is no anterior activity. If you want to be blind to some data, then fine but do not claim there is nothing to see, when you don’t (want to) watch, c) you do seem to analyze the anterior electrodes. 164, 208: “branvision”

Results:

A number of notation errors “p” and “d” where punctuation signs are missing. Line 341. What is “vMT”?

Discussion:

The argument about language processing is misleading, as many studies reported frontal ERP activity for visual tasks (e.g. on the frontal selection positivity, FSP), while language tasks are associated with many more areas than just the frontal scalp. The FSP falls also in the time range where the authors find a difference between which falls in the time range were the authors report a more positive ERP for the control condition. Please discuss that.

Round 2

Reviewer 2 Report

I actually meant Talsma, Wijers, Klaver & Mulder (2001), instead of Talsma & Kok (2001), because they used similar polygons. It could be argued that the contralateral polygon in Talsma et al. 2001 present an active (2 items to be memorized) or inactive (1 item to be memorized) context/background. The status of the other item has only limited effect on maintenance (Klaver, Talsma, Wijers & Mulder 1999), but some effect on the P1 latency and target detection (Talsma, Wijers, Klaver & Mulder 2001). Even for such complex objects as polygons. This has direct relevance to the rationale that the authors use in line 211-214: “… evidence that recall performance may depend on the task-relevant feature to memorize [59], and thus we attempted to keep the same feature domain of the task-relevant polygons”. In the study reported here, the number of objects is higher (4 items). It would be interesting to see the authors view on comparing that with the findings in their study, in the context of “vWM is tolerant to visual item loading and configuration change”. The discussion on the N2pc (line 402) does not make sense, because the N2pc was not recorded. Typo line 125: “Talsma” instead of “Talsman” Typo line 148: “polygon” instead of “polygong” Separate analyses for match/mismatch trials should be reported, even when no significance was reached.

A number of errors claimed to be corrected were not corrected:
- the errors in lines 178 and 222 “branvision” were not corrected.

Line 355 vMT was not corrected

Author Response

This manuscript is a resubmission of an earlier submission. The following is a list of the peer review reports and author responses from that submission.

Round 1

Reviewer 1 Report

This manuscript presents an EEG experiment supposedly investigating the effect of visual interference on visual working memory. In a change detection task, for which participants had to memorize four polygons, the level of “visual noise” was manipulated by adding background shapes linking the memory items (either a square or a polygon). Performance under these conditions was compared to a condition without any background stimulus. There was no effect on behavioural performance. The P2 over occipital cortex following the test array was found to be larger in the conditions with background shapes.

I have several concerns related to virtually all aspects of the study (background, rationale, design, data analysis and interpretation, conclusions) and I list some of the most important ones below.

1) The question addressed with this study is unclear to me – the only thing that seems clear is that the authors were trying to find an effect (any effect) of “visual noise”. That is too broad and by no means a new area of research, which could warrant such an exploratory approach. The manuscript remains very vague in this regard, for instance the question is specified as follows: “[…] whether the visual noise has an effect on correct-recall ratios as well as brain activity during a visual working memory task” (abstract; - this is really just a description of the measures); “To explore this issue [the effects of visual noise on VWM], we studied the behavioural responses and ERP components […]; “[…] we will observe both behavioural differences related to recall, and changes in the ERP waveform” (- Again, merely a description of the measures and the “hypothesis” that there will be some changes.)

2) It seems like the authors are not very familiar with the field, the extant literature and how their study fits into that. I am missing a proper introduction to interference, for instance its different forms (e.g., sensory interference or interference from other items residing in memory) and current opinions on its role for visual working memory. This is a pretty active area of research.

3) I would not use the term “visual noise”, as this term is reserved for something else in vision science. The task-irrelevant background information used in the present study could rather be seen as distractor objects, and their variation (square vs. polygon) as a manipulation of target-distractor similarity. Thus, different processes might come into play here (e.g., object individuation, filtering at encoding, interference during maintenance if the task-irrelevant background shape was encoded), but this is certainly not (only) a manipulation of “sensory interference”, which the authors apparently set out to investigate (e.g., page 2).

4) Behavioural performance was at chance in the task (55%, 54% and 53% in the different conditions). For one, this is a straightforward explanation for why there were no behavioural effects that is not even mentioned in the manuscript. More importantly, I do not see how any conclusions with respect to the role of interference can be drawn based on these data, when participants were clearly not able to perform above chance. This does not only affect the behavioural but also the electrophysiological data, as incorrect trials were not removed from the analyses.

5) Related to issues raised in my previous comments, the one effect that is observed in this study – increased P2 amplitudes over occipital cortex following the onset of the test array in the “square” and “irregular” conditions as compared to the control condition – could reflect a number of processes, not even necessarily processes related to visual memory. Assuming that the “noise” was not only present in the memory array but also in the test array (which is not clear to me), the origin of the P2 effect could also very well be sensory in nature, contrary to what the authors state in the discussion (page 15).

Author Response

Response to Reviewer 1   First, a native english speaker review the paper. He gave us feedback to correct setences in the manuscript. Based on comments, we made a major revision of the paper.    1) The question addressed with this study is unclear to me – the only thing that seems clear is that the authors were trying to find an effect (any effect) of “visual noise”. That is too broad and by no means a new area of research, which could warrant such an exploratory approach. The manuscript remains very vague in this regard, for instance the question is specified as follows: “[…] whether the visual noise has an effect on correct-recall ratios as well as brain activity during a visual working memory task” (abstract; - this is really just a description of the measures); “To explore this issue [the effects of visual noise on VWM], we studied the behavioural responses and ERP components […]; “[…] we will observe both behavioural differences related to recall, and changes in the ERP waveform” (- Again, merely a description of the measures and the “hypothesis” that there will be some changes.)
We thank this observation.
We have clarified and made explicit our goal, the means of doing it through the task paradigm, and the expected results. You can read it in between lines 23 and 33; and between lines 65 and 88.
  2) It seems like the authors are not very familiar with the field, the extant literature and how their study fits into that. I am missing a proper introduction to interference, for instance its different forms (e.g., sensory interference or interference from other items residing in memory) and current opinions on its role for visual working memory. This is a pretty active area of research.
We thank for this observation.
We added the references needed for state our hypothesis and our choice of CDT. We related it with the brain processes that we think will be involved in such a way that we tried to make clear the rationale of our study in the research field. You can read it between lines 79 and 88.
  3) I would not use the term “visual noise”, as this term is reserved for something else in vision science. The task-irrelevant background information used in the present study could rather be seen as distractor objects, and their variation (square vs. polygon) as a manipulation of target-distractor similarity. Thus, different processes might come into play here (e.g., object individuation, filtering at encoding, interference during maintenance if the task-irrelevant background shape was encoded), but this is certainly not (only) a manipulation of “sensory interference”, which the authors apparently set out to investigate (e.g., page 2).
We thank for this suggestion. All through the document, we have changed the "noise" word for the "interference" or "sensory interference" description of our intended concept.
  4) Behavioural performance was at chance in the task (55%, 54% and 53% in the different conditions). For one, this is a straightforward explanation for why there were no behavioural effects that is not even mentioned in the manuscript. More importantly, I do not see how any conclusions with respect to the role of interference can be drawn based on these data, when participants were clearly not able to perform above chance. This does not only affect the behavioural but also the electrophysiological data, as incorrect trials were not removed from the analyses.
We appreciate this observation. We now made clear in the document that we only analyzed the correct trials for ERP results.
On the other hand, we have written some ideas regarding the behavioral results with respect our hypothesis and the understanding of vWM process and the brain related functioning. This ideas appear in the discussion section, between the lines 314 and 350.
  5) Related to issues raised in my previous comments, the one effect that is observed in this study – increased P2 amplitudes over occipital cortex following the onset of the test array in the “square” and “irregular” conditions as compared to the control condition – could reflect a number of processes, not even necessarily processes related to visual memory. Assuming that the “noise” was not only present in the memory array but also in the test array (which is not clear to me), the origin of the P2 effect could also very well be sensory in nature, contrary to what the authors state in the discussion (page 15).
We appreciate this observation. We made the amendments for solving this lack of argumentation between our brain functioning and vWM hypothesis with respect to our memory task. This is in the discussion section of the document. You can read it between the lines 292 and 313.

Reviewer 2 Report

The authors investigate the effect of noisy and noise-free backgrounds in a modified Change Detection Test on behavioral results, using the recall process, and electrophysiological results using event-related potentials in the EEG. The authors' hypothesis is, that the presence of visual noise influence on the visual working memory will result in changing the behavioral results as well as in the ERG waveforms.

Overall comments: I think the manuscript would benefit from proof-reading by a native speaker. Please use a consistent reference to figures (i. e. Fig. vs. Figure).

Abstract:

Line 3: Could you maybe add the types of the backgrounds used

Introduction:

Line 24: … have been stated?

Line 26: … there exists a possibility …

Line 39: Any Reference for N1, P1, and P3?

Line 43: “raised” instead of “heightened”?

Line 51: .. items, the reported a P2 … Did you mean “they” reported?

Material and Methods:

Lines 73,74: Please do not report the age range using SEM. The standard error of means (SEM) is used for describing the variability of means of a random sampling process. The standard deviation would be the appropriate descriptive statistics. Please also add years as a unit.

Lines 85,86: I think it is enough to report brightness values as rounded integer values since they are highly variable, affected already by a slight change in the angle of the lux meter. Additionally, I would not report the standard deviations in the measurements, unless you give the number of the repeated measurements.

Line 90: I think you mean the International 10-20 system

Lines 94,95: Could you add the version of PyCoder used in the study?

Lines 97: Could you add the versions of EEGLAB and ERPLab?

Line 102: Did you mean .., while the degree of complexity follows the recommendations …?

Line 105: Just out of curiosity: Why is the interval for the test array shorter than the intervals for memory array and retention?

Lines 129-132: This was already mentioned before and may be skipped.

Line 159: Please use the same style for all references.

Lines 160,161: Please rephrase this sentence. There are word doublings.

Line 168: Could you provide a reference for the settings of the filters for blink-removal?

Results:

Lines 204-207: Does a median response of about 0.5 indicate that the correct response was selected only by chance?

Lines 217-223: It is not surprising that the zone of recording (frontal, occipital, parietal) has a statistically significant effect on the amplitude of the P2 response. I would suggest excluding the zone as a factor and analyze the different zone separately since the zone does not provide any new information.

Lines 230-243: Same as in the comment before. I don’t see any new information here. It is not surprising that any stimulus will show different P2 amplitudes in different areas.

Lines 245-259: Same as before.

Lines 261-267: Please correct me if I’m wrong, but again: It is not surprising that the variability in the P2 amplitude is mainly caused by the recording site.

I would also not expect the reaction time as a significant predictor since you have not found a statistically significant difference between the reaction times. What about the accuracy? You did not mention it, even if you have included it as a factor.

Discussion:

I’m not quite sure if the increased P2 amplitude in the control and irregular blocks may not be caused simply by the additional shapes in the stimulus, which creates additional contrast. Could you discuss such an effect?

Author Response

Response to reviwer 2. A native english speaker review the document. He gave us feedback for correcting sentences in the document. Based on your comments, we review the manuscript.   Line 24: … have been stated?
We've rephrased this line during the revision.
  Line 26: … there exists a possibility … 
We rephrased this line.
  Line 39: Any Reference for N1, P1, and P3?
We added references for the three components.   Line 43: “raised” instead of “heightened”?
We modyfied the sentence
  Line 51: .. items, the reported a P2 … Did you mean “they” reported?
This is correct, the typographical error was corrected.
  Lines 73,74: Please do not report the age range using SEM. The standard error of means (SEM) is used for describing the variability of means of a random sampling process. The standard deviation would be the appropriate descriptive statistics. Please also add years as a unit.   Thank you for this observation, we've corrected these lines to report standard deviation instead. We also included years as the unit. Lines 85,86: I think it is enough to report brightness values as rounded integer values since they are highly variable, affected already by a slight change in the angle of the lux meter. Additionally, I would not report the standard deviations in the measurements, unless you give the number of the repeated measurements. We've made the suggested changes, rounded brightness measurements to integers and removed standard deviations.   Line 90: I think you mean the International 10-20 system This is correct, we've corrected the typographical error.   Lines 94,95: Could you add the version of PyCoder used in the study? We added the version of Pycorder, which is 1.0.4.   Lines 97: Could you add the versions of EEGLAB and ERPLab? We added the versions for both programs, which are EEGLAB version 13.6.5  and  ERPlab version 5.0   Line 102: Did you mean .., while the degree of complexity follows the recommendations …? Thank you for the observation, we corrected the sentence.   Line 105: Just out of curiosity: Why is the interval for the test array shorter than the intervals for memory array and retention? This interval was cut down in length to reduce the overall duration of the experiment. During the pilot stage we realized that most subjects responded well before 1200 miliseconds. In the final experiment, the average response time in this interval was 857 miliseconds.   Lines 129-132: This was already mentioned before and may be skipped. These lines were skipped as suggested.

Round 2

Reviewer 1 Report

I appreciate that the authors have made some changes to the manuscript, but unfortunately, none of my previous comments have been adequately addressed:

The research question and hypotheses are still not sufficiently precise. Some literature has been added, but this does not improve clarity at all. There is still no proper introduction to interference, no concise description of the current state of research and how this study would contribute to the field, no clarification of what kind of interference is studied here (in fact, some literature refers to interference from other items in memory, and not from sensory input). The introduction (as well as the discussion) could strongly benefit from some streamlining of the content and from a clearer structure. Contrary to what the authors state in their response, “visual noise” has not been changed to “(sensory) interference” throughout the manuscript. There are plenty of instances, in which (visual) noise is still used – this just adds some confusion, because neither term is used consistently. and 5) I do not see any substantial changes to the discussion that attempt to address the issues I raised in these comments.

I am afraid that I do not see how this study makes any contribution to our understanding of visual working memory. As I described in my first review, the P2 effect that is observed could be related to a number of processes, not even necessarily visual memory. This should ideally be addressed in further experiments with modified designs, but at the very least be adequately and clearly discussed. Unfortunately, the authors have not made the effort to do either.